# Survival Impact of Robotic-Assisted Laparoscopy (RAL) vs. Conventional Laparoscopy (LPS) in the Treatment of Endometrial Cancer

**DOI:** 10.3390/cancers17030435

**Published:** 2025-01-27

**Authors:** Vanesa Delso, Rafael Sánchez-del Hoyo, Lucía Sánchez-Barderas, Myriam Gracia, Laura Baquedano, María A. Martínez-Maestre, María Fasero, Pluvio J. Coronado

**Affiliations:** 1Women’s Health Institute, Hospital Clínico San Carlos, IdISSC–Instituto de Investigación Sanitaria del Hospital Clínico San Carlos de Madrid, School of Medicine Complutense University, 28040 Madrid, Spain; sanchezbarderas16@gmail.com (L.S.-B.);; 2Research Methodological Support Unit and Preventive Department, Hospital Clínico San Carlos, IdISSC–Instituto de Investigación Sanitaria del Hospital Clínico San Carlos de Madrid, 28040 Madrid, Spain; rafaelsanchezdelhoyo@gmail.com; 3Gynecologic Oncology Unit, La Paz University Hospital—IdiPAZ, 28046 Madrid, Spain; dra_gracia@hotmail.com; 4Department of Obstetrics and Gynecology, Miguel Servet Hospital, 50009 Zaragoza, Spain; lbaquedanome@hotmail.com; 5Department of Obstetrics and Gynecology, Hospital Virgen del Rocío, 41013 Sevilla, Spain; martinez.maestre@hotmail.com; 6Menopause Unit, Hospital Sanitas la Zarzuela, Corofas Menopause, 28001 Madrid, Spain; mfaserol@gmail.com

**Keywords:** endometrial carcinoma, survival, robotic-assisted laparoscopy, complex surgical, laparoscopy, obesity, morbidity

## Abstract

We conducted a retrospective study of 723 women with early-stage endometrial cancer who underwent minimally invasive surgery. Of these, 468 had laparoscopic surgery (LPS), and 255 had robotic-assisted laparoscopy (RAL). After matching patients by key factors, no significant differences were found in disease-free survival (DFS), overall survival (OS), or specific survival (SS) between the two surgical approaches. This suggests that the choice of surgery does not affect survival outcomes in women with EC.

## 1. Introduction

The second most common gynecological cancer worldwide is uterine cancer [1]. Endometrial cancer (EC) is the most prevalent cancer of the female reproductive organs in the United States [2,3]. More than 80% of endometrial cancers are endometrioid [4], which is typically associated with low-grade tumors and a good prognosis [5]. For FIGO stage I EC, the 5-year disease-free survival (DFS) rate ranges from 80 to 95% [6]. Minimally invasive surgery (MIS) is considered the main approach in these tumors, although some studies have questioned the safety of MIS for endometrial cancer [7,8]. MIS revealed equivalence of survival outcomes in EC when the open approach and laparoscopic approach (LPS) were compared, according to the Gynecologic Oncology Group Study LAP2 trial and LACE trial [9,10]. Similar results were obtained in other retrospective studies, such as the MISOPEC study [11], where equivalent oncologic outcomes between robotic-assisted laparoscopy (RAL) and LPS overall survival were observed [11]. However, according to Jørgensen et al., LPS and RAL were associated with longer endometrial cancer survival than the open approach [12]. The safety of MIS is consistent, so the current European guidelines have adopted LPS and RAL as the gold standard in endometrial cancer surgery [13]. The advantages of the RAL have been considered a promising alternative to LPS not only because of its greater dexterity, visualization, and accuracy but also because of its better perioperative outcomes [7].

Clinical trials comparing LPS and RAL are limited. Due to the scarcity of evidence regarding these minimally invasive surgical approaches in gynecological cancers, the aim of our study was to evaluate the impact of both surgical techniques on disease-free survival and overall survival in women diagnosed with EC.

## 2. Materials and Methods

### 2.1. Study Design and Participants

We conducted a multicenter retrospective cohort study of women diagnosed with EC between 2005 and 2022 at four tertiary Spanish medical centers. The majority of these centers began using minimally invasive surgical (MIS) approaches for EC treatment in 2005 with LPS, followed by RAL in 2007. All cases of preoperative early-stage EC whose primary treatment was at least hysterectomy and bilateral salpingo-oophorectomy were included. We excluded patients whose primary treatment was radio- or chemotherapy, uterine sarcomas, and cases with synchronic ovarian cancer. According to current guidelines, pelvic or pelvic and para-aortic lymph node dissections were performed [13,14]. RAL and LPS procedures were performed by the same gynecologic oncologic team. All cases of RAL were performed with the Da Vinci System standard and Xi, and the election of the approach depended on the availability of the robotic system (only once per week for gynecology). No uterine manipulators were used in most cases; instead, we used a vaginal device made at our institution that is similar to the AppleTM vaginal probes. Prophylactic antibiotics and low molecular weight heparin were always administered preoperatively. Intravenous fluids were usually maintained in the first 12 h after surgery until patients tolerated oral fluids. When the patient tolerated oral fluids, the Foley catheter was removed. If the patients tolerated a regular diet, had stable vital signs, demonstrated the ability to ambulate independently, and controlled pain with oral analgesics, they were discharged. Two pathologists with expertise in gynecologic tumors analyzed the surgical pieces. According to the guidelines of the European Society of Gynecologic Oncology and the Spanish Society of Gynecology and Obstetrics, at the time of diagnosis, the adjuvant treatment was administered [13,14] in agreement with the Institutional Tumor Board, consisting of external beam radiotherapy (EBRT), brachytherapy, and/or chemotherapy depending on each case [13,14]. The patient’s status at the last visit and the presence of recurrences were recorded during follow-up visits performed at least every 4–6 months.

### 2.2. Propensity Score Model

To minimize heterogeneity in this retrospective study, we employed a statistical model using propensity score matching. This approach serves as the best approximation to a randomized controlled trial, helping to reduce selection bias and account for confounding factors in patient selection. In selecting the variables for the model, we focused on factors that could influence the choice of surgical approach, the use of adjuvant treatment, or patient survival. These factors included age at diagnosis, body mass index (BMI), pre-surgical comorbidities, family history of cancer, American Society of Anesthesiologists (ASA) score, histological subtype, histological grade, myometrial invasion, lymphovascular space invasion (LVSI), and FIGO stage. The flow chart illustrating the selection of matched pairs and the reasons for withdrawal is shown in Figure 1. A total of 723 patients, corresponding to 242 matched pairs, were included in the final analysis (Figure 1).

### 2.3. Statistical Analysis

A stratified analysis was conducted comparing RAL and LPS. Qualitative variables are presented as frequency distributions. Quantitative variables are summarized using the mean and standard deviation (SD), while those with asymmetric distributions are described by the median and interquartile range (IQR). Discrete variables are presented as absolute frequencies and relative percentages. The propensity score matching technique was performed with the variables age at diagnosis, body mass index (BMI), pre-surgical comorbidities, American Society of Anesthesiologists (ASA) score, histological subtype, histological grade, myometrial invasion, lymphovascular space invasion, and FIGO stage, to homogenize the RAL group and the LPS group. To study the association between qualitative variables, a chi-squared test or Fisher’s exact test was conducted when appropriate. Comparisons between groups were studied by Student’s *t*-test with normal distribution. Mann–Whitney tests were used for non-parametrical variables.

The relationship between the surgical approach and mortality or recurrence was further explored through survival analysis using Kaplan–Meier curves. The log-rank test was applied to compare survival functions. For survival analysis, Cox regression was used to assess the association between the study groups and disease-free survival (DFS), overall survival (OS), or tumor-specific survival (SS). Multivariate analysis with Cox’s proportional hazards model was employed to adjust the hazard ratio (HR) with a 95% confidence interval (CI). Data processing and analysis were performed using IBM SPSS Statistics v.26 and R statistical software v. 4.3.1.

In accordance with the journal’s guidelines, we will make our data available for independent analysis by a team selected by the Editorial Board should additional analysis or reproducibility of the study in other centers be requested.

## 3. Results

### 3.1. Whole Sample

A total of 723 women with primary early-stage endometrial cancer (EC) who underwent minimally invasive surgical staging as primary treatment were included (Table 1). LPS was performed in 468 (64.7%) cases, while 255 (35.3%) patients underwent RAL. The general characteristics of both study groups are presented in Table 1. Compared to LPS, women who underwent RAL surgery had higher body mass indices (BMIs) (*p* = 0.029), more comorbidities (*p* = 0.005), and worse ASA scores (*p* = 0.005). Regarding histopathological features, patients in the RAL group had a higher rate of lymphovascular space invasion (LVSI) (*p* = 0.005). Additionally, adjuvant chemo-radiotherapy was more frequently indicated in the RAL group (*p* < 0.001).

### 3.2. Propensity Score Analysis

We included 241 matched pairs (482 women) of patients with EC in the study. Patient characteristics, preoperative findings, pathology results, and treatment details are presented in Table 2. Both surgical groups were similar across all variables.

After a mean follow-up of 65.5 ± 46.7 months, 74 women (15.4%) experienced a relapse, and 89 patients (18.5%) died. Among the deceased, 43 cases (8.9%) were attributed to disease-related deaths, while 46 cases (10%) were due to other causes. The 5-year disease-free survival (DFS) rate was 84.6% in both the LPS and RAL groups (HR 1.0, 95% CI 0.634–1.577; *p* = 0.999). The 5-year overall survival (OS) rate was 81.7% in the LPS group and 81.3% in the RAL group (HR 0.907, 95% CI 0.707–1.662; *p* = 0.711). The 5-year specific survival rate related to endometrial cancer (EC) was 92.9% in the LPS group and 89.2% in the RAL group (HR 0.154, 95% CI 0.799–2.716; *p* = 0.215) (Figure 2).

## 4. Discussion

Our study found that the choice of a minimally invasive approach did not impact oncologic outcomes in patients with endometrial cancer. Propensity score analysis revealed no significant differences in disease-specific survival, disease-free survival, or overall survival. Several articles have been published recently that support our results [15,16].

Minimally invasive surgery (MIS) has changed gynecologic oncology [17]. Robotic-assisted laparoscopy (RAL), besides the standard LPS, portrays a commonly applied surgical alternative approach in EC staging [18]. Despite this, concerns regarding the safety of MIS in gynecological cancers persist.

Minimally invasive surgery has been shown to be the preferred approach for treating endometrial cancer in two prospective randomized studies [19,20]. Additionally, the current European guidelines designate the minimally invasive approach as the method of choice in the management of endometrial cancer [13]. The minimally invasive approach is characterized by decreased intraoperative and postoperative complications and bleeding, reduced wound complications and pain, and shorter hospital stay and recovery [21]. Although several studies have confirmed the superiority of MIS on oncologic outcomes, they have failed to demonstrate an impact on survival [7,11,18].

Both RAL and LPS appear to be oncologically safe for the treatment of endometrial cancer, with an improvement in quality of life observed during the first 6 months [22,23,24], perioperative complications not increased, and postoperative complications reduced [11].

RAL safety and feasibility have been thoroughly investigated, with key advantages over standard LPS, including the 3-dimensional view, enhanced dexterity of the robotic arms, and a shorter learning curve [25]. Some studies have compared both approaches using objective measurement tools, with shorter knotting and suturing times observed in robotic surgery compared to laparoscopy [26]. This aspect is particularly important in overcoming the technical challenges posed by complex patients, such as those with endometrial cancer (EC) [27].

RAL offers a significant advantage in challenging surgical cases, such as those involving obese or elderly patients, both of which are factors commonly associated with endometrial cancer. The enhanced visualization and improved dexterity for suturing in confined spaces, such as those encountered in obese patients, provide a clear benefit of RAL over LPS [26]. RAL has shown advantages in treating obese women with endometrial cancer, including reduced blood loss and lower rates of conversion to laparotomy [28]. While some reports suggest that RAL may outperform LPS in terms of perioperative outcomes [29], the literature, though extensively studied, remains controversial on this issue [11]. Additionally, RAL is associated with fewer postoperative complications, shorter hospital stays, and reduced complexity of the approach, which greatly benefits frail patients, such as the elderly [29].

Another important consideration is the need for protective maneuvers to prevent tumor spillage, including tumor rupture or morcellation [13,30]. While robotic-assisted laparoscopy (RAL) offers advantages such as 3D visualization and the enhanced dexterity of robotic arms, features that allow for greater control and precision during movements and could, theoretically, reduce the risk of secondary tumor dissemination during tissue extraction, there is currently insufficient evidence to demonstrate significant differences between RAL and conventional laparoscopy (LPS) in this regard [26]. Similarly, although the uterine manipulator is considered useful in laparoscopic surgery, its use remains a controversial topic due to the potential risks of uterine perforation, which could increase the likelihood of tumor dissemination [31,32,33]. In robotic-assisted laparoscopic surgery (RAL), the use of a uterine manipulator is often not necessary, given the advantages that the robotic technique offers, such as enhanced visualization and precision. However, this is not the case in traditional laparoscopic approaches, where many centers continue to rely on the manipulator.

In the context of our study, the uterine manipulator was employed in a limited number of centers. While some studies suggest that the use of the manipulator may negatively impact oncologic outcomes [32], there is currently no solid evidence or formal recommendation in the main guidelines for the management of endometrial cancer to discontinue its use. Although it was used in a few centers in our study, this could represent a potential limitation, as it may introduce bias if its involvement in survival outcomes is proven.

A limitation of this study is its retrospective multicenter design. However, we addressed potential confounding factors through propensity score matching analysis. Furthermore, the large sample size, real-world data, and long-term follow-up provided sufficient statistical power, reinforcing the robustness of our results.

It is crucial, however, that when new techniques are adopted in the treatment of oncology patients, their oncological efficacy and safety should be rigorously established rather than assumed based on first principles.

## 5. Conclusions

This propensity score analysis suggests that MIS is safe in the management of endometrial cancer, with similar long-term oncological outcomes between RAL and LPS. RAL is comparable to LPS regarding OS, SS and DFS.

## Figures and Tables

**Figure 1 cancers-17-00435-f001:**
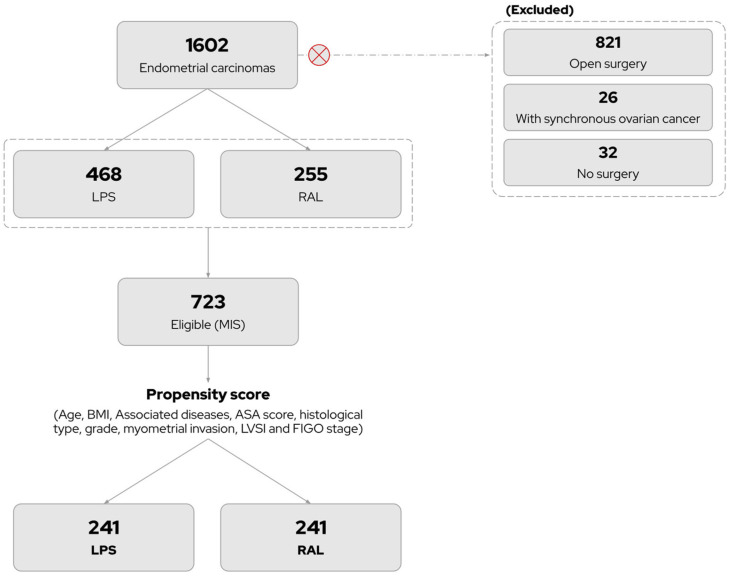
A flowchart indicating the selection of propensity score. MIS; minimally invasive surgery. LPS; conventional laparoscopy. RAL; robotic-assisted laparoscopy.

**Figure 2 cancers-17-00435-f002:**
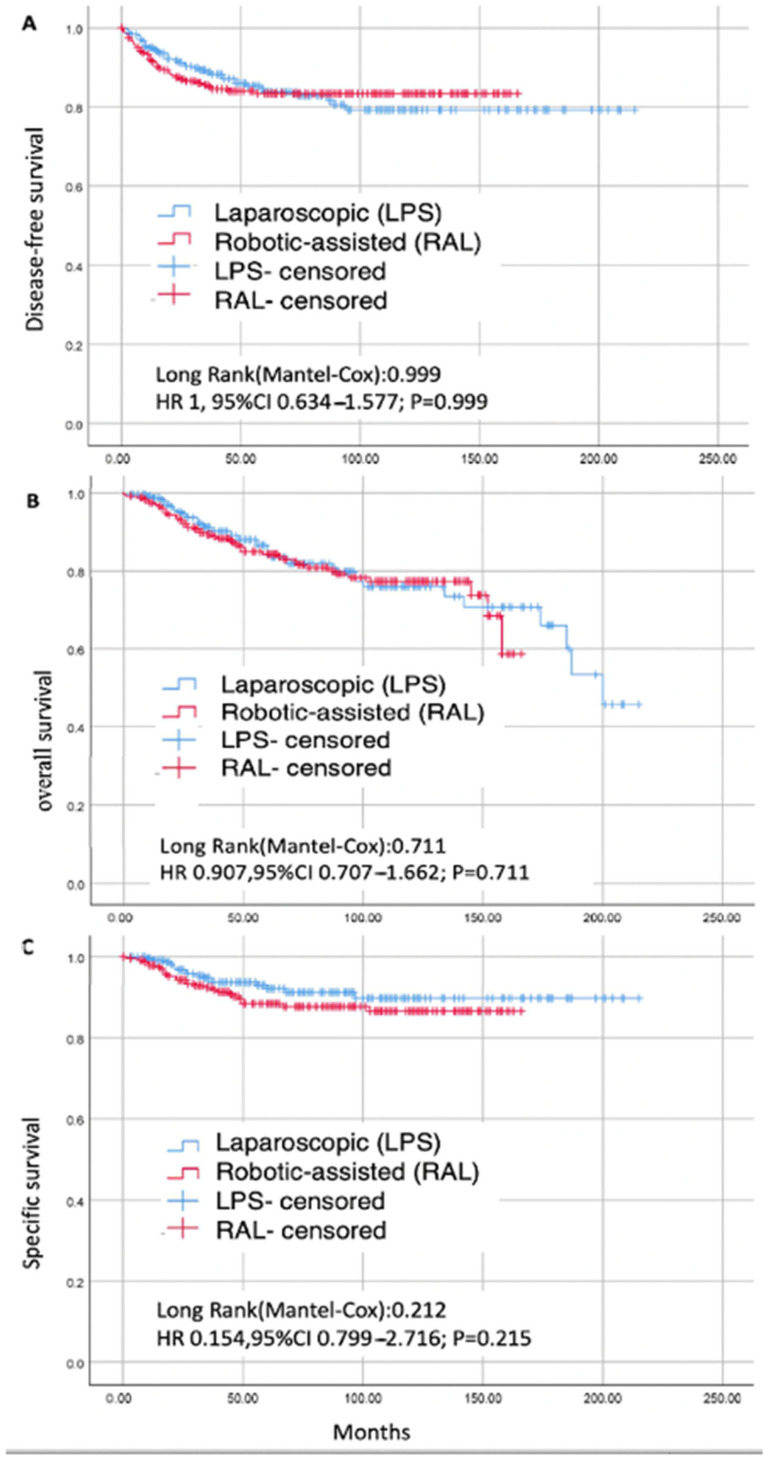
Kaplan–Meier curves were generated for the matched surgical groups. The hazard ratios, 95% confidence intervals, and corresponding *p*-values were calculated using Cox’s proportional hazards model. Panel (**A**) shows the disease-free survival curve, panel (**B**) presents the overall survival curve, and panel (**C**) depicts the specific survival curve.

**Table 1 cancers-17-00435-t001:** Demographic and pathological characteristics of all patients in the cohort before matching (*n* = 723) with endometrial cancer (EC).

Variable	LPS	RAL	*p*-Value
	N = 468	N = 255	
Age (years)	64.86 ± 10.7	65.52 ± 10.79	0.433
BMI (kg/m^2^)	28.44 ± 5.6	29.42 ± 5.9	0.029
Associated diseases	50 (11.6%)	47 (19.4%)	0.005
American Society of			
Anesthesiologists (ASA)			
II	367 (82.7%)	185 (73.7%)	0.005
III–IV	77 (17.3%)	66 (26.3%)	
Family history of cancer	176 (37.8%)	70 (27.6%)	0.005
Year from menopause	14 ± 11.1	14 ± 11.5	0.718
Parity	2 ± 1.7	2 ± 1.9	0.088
Histologic subtype			
Endometrioid	391 (83.9%)	211 (83.7%)	0.951
No endometrioid	75 (16.1%)	41 (16.3%)	
Histological grade			
G1–G2	374 (80.6%)	190 (75.4%)	0.104
G3	90 (19.4%)	62 (24.6%)	
Myometrial invasion			
<50%	244 (53.0%)	149 (59.8%)	0.082
>50%	216 (47.0%)	100 (40.2%)	
LVSI			
No	353 (78.4%)	216 (87.1%)	0.005
Yes	97 (21.6%)	32 (12.9%)	
FIGO (2009) stage			
I–II	402 (87.4%)	211 (84.7%)	0.324
III–IV	58 (12.6%)	38 (15.3%)	
Lymphadenectomy pelvic			
No	184 (39.3%)	128 (50.2%)	0.005
Yes	284 (60.7%)	127 (49.8%)	
Lymphadenectomy para-aortic			
No	340 (72.8%)	212 (83.1%)	0.002
Yes	127 (27.2%)	43 (16.9%)	
Adjuvant therapies			
No	192 (41%)	116 (45.5%)	
Radiotherapy	230 (49.1%)	97 (38%)	<0.001
Chemotherapy	6 (1.3%)	4 (1.6%)	
Radio-chemotherapy	40 (8.5%)	38 (14.9%)	
Length of follow-up (months)	59.80 ± 50	68.7 ± 47.3	0.34

Data are presented as mean ± standard deviation and frequencies (percentages). BMI: body mass index. LVSI: lymphovascular space invasion. 1 Includes cardiovascular diseases, thromboembolic diseases, chronic pulmonary diseases, and liver diseases.

**Table 2 cancers-17-00435-t002:** Demographics and pathology results of patients after propensity score matching (n = 482).

Variable	LPS	RAL	*p*-Value
	N = 241	N = 241	
Age (years)	65.0 ± 10.29	65.0 ± 10.65	0.781
BMI (kg/m^2^)	28.6 ± 5.89	29.40 ± 5.88	0.584
Associated disease	29 (12.5%)	44 (19.0%)	0.056
American Society of			
Anesthesiologists (ASA)			
I–II	187 (77.6%)	179 (74.3%)	0.394
III–IV	54 (22.4%)	62 (25.7%)	
Family history of cancer	62 (25.7%)	68 (28.2%)	0.538
Year from menopause	12.5 ± 11.6	14 ± 11.5	0.748
Parity	2 ± 1.8	2 ± 1.9	0.297
Histologic subtype			
Endometrioid	203 (84.2%)	205 (85.1%)	0.8
No endometrioid	38 (15.8%)	36 (14.9%)	
Histological grade			
G1–G2	181 (75.1%)	180 (74.7%)	0.916
G3	60 (24.9%)	61 (25.3%)	
Myometrial invasion			
<50%	146 (60.6%)	145 (60.2%)	0.926
>50%	95 (39.4%)	96 (39.8%)	
LVSI			
No	209 (86.7%)	210 (87.1%)	0.893
Yes	32 (13.3%)	31 (12.9%)	
FIGO (2009) stage			
I–II	208 (86.3%)	204 (84.6%)	0.605
III–IV	33 (13.7%)	37 (15.4%)	
Lymphadenectomy pelvic			
No	104 (43.0%)	114 (47.3%)	0.36
Yes	137 (56.8%)	127 (52.7%)	
Lymphadenectomy para-aortic			
No	182 (75.5%)	198 (82.2%)	0.074
Yes	59 (24.5%)	43 (17.8%)	
Adjuvant therapies			
No	105 (43.4%)	106 (43.8%)	
Radiotherapy	108 (44.6%)	96 (39.7%)	-
Chemotherapy	4 (1.7%)	4 (1.7%)	
Radio-chemotherapy	25 (10.3%)	36 (14.9%)	
Length of follow-up (months)	59.10 ± 50.91	71.5 ± 46.58	0.221

Data are presented as mean ± standard deviation and frequencies (percentages). BMI: body mass index. LVSI: lymphovascular space invasion. 1 Includes cardiovascular diseases, thromboembolic diseases, chronic pulmonary diseases, and liver diseases.

## Data Availability

The data presented in this study are available on request from the corresponding author. The data are not publicly available because they contain information that could compromise the privacy of patients.

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
