# Peer review of "Survival Impact of Robotic-Assisted Laparoscopy (RAL) vs. Conventional Laparoscopy (LPS) in the Treatment of Endometrial Cancer"

_cancers, 2025, doi:10.3390/cancers17030435_

Round 1

Reviewer 1 Report

Comments and Suggestions for Authors

This is a well written manuscript that asks an important question.

I have a few questions/suggestions for the authors.

1) The numbers here are large but maybe not large enough to detect a difference.  Some power analysis in the discussion might be helpful as to how many patients would be required to detect a small difference.

2) Discussion of the reason for the differences in the RAL a LPS group would be helpful.  Why are the RAL patients heavier?  Were there selection biases?

Comments on the Quality of English Language

The English is fine

Author Response

Comments 1) The numbers here are large but maybe not large enough to detect a difference.  Some power analysis in the discussion might be helpful as to how many patients would be required to detect a small difference.

Response: Thank you for your insightful comment. Based on the data we have, the statistical analysis used, and the sample size of 241 patients per arm with a 6-year follow-up, our results appear robust. While increasing the sample size may improve the precision of the estimates, it is unlikely to significantly alter the outcomes given the small effect sizes observed (HR close to 1.0). In fact, detecting smaller differences would likely require an even larger sample, especially considering the absence of a substantial difference between the groups.

Comments 2) Discussion of the reason for the differences in the RAL a LPS group would be helpful.  Why are the RAL patients heavier?  Were there selection biases?

Response:  Thank you for your comment and for raising the question. The patients were not specifically selected based on their characteristics for the choice of surgical approach (RAL vs LPS). However, robotic surgery was prioritized for more complex cases when available. As a result, the descriptive sample prior to statistical analysis included patients with higher ASA scores, more comorbidities, and a higher BMI. A propensity score analysis was performed, which helps account for any potential biases in the assignment of patients to either group. This statistical method adjusts for differences in baseline characteristics (such as BMI and other relevant factors) between the two groups, ensuring that if there had been any biases in group allocation, they would have been corrected. By performing propensity score matching, the results may approximate those of a randomized controlled trial (RCT). Therefore, the propensity score analysis would have eliminated any potential confounding factors and ensured that the comparison between the two groups (RAL and LPS) was valid and unbiased.

Reviewer 2 Report

Comments and Suggestions for Authors

Dear authors, 

congratulations for this relevant work. 

Please find my comments as follows: 

- Line 42 and Line 177 / 198: I would recommend to clearly focus on EC and not mix with lietarure on cervical cancer. 

- Line 211: for staging or for surgical treatment?!

- Table S1 needs some language revisions (Spanish words...)

- Did you exclude the FIGO stage III and IV cases? In the results section you are reporting about early stage disease.

- Why is there a significance in chemotherapy while the number re the same in the table S1?

- Line 201: please make clear what this means? Why drawback?

- the possible advantages of RAS could be discussed in more detail: obesity, elderly, ICG use, etc. 

- if you discuss spillage, you also could add the aspect of tissue extraction and discuss differences between laparoscopy and RAS. 

Kind regards. 

Comments on the Quality of English Language

Some Spanish words in table S1 require translation

Author Response

Please find my comments as follows: 

- Line 42 and Line 177 / 198: I would recommend to clearly focus on EC and not mix with lietarure on cervical cancer. 

R: Thank you for your valuable suggestion. I appreciate your feedback and will make sure to focus more clearly on endometrial cancer (EC) in the relevant sections, avoiding any confusion with the literature on cervical cancer.

- Line 211: for staging or for surgical treatment?!

R: Thank you very much for your thoughtful comment. Staging in endometrial cancer is indeed surgical, making it both a staging and treatment procedure. I will make sure to clarify that it refers to a surgical treatment to ensure it is clearer in the text.

"Both robotic and laparoscopic approaches are safe options for treating endometrial cancer in its apparent early stages."

- Table S1 needs some language revisions (Spanish words...)

R: Thank you for your feedback. I appreciate your observation.

- Did you exclude the FIGO stage III and IV cases? In the results section you are reporting about early stage disease.

R: Thank you for your question. We included patients with early-stage disease who underwent surgical staging, provided their primary treatment was minimally invasive surgery. Additionally, patients whose FIGO stage changed after staging surgery were not excluded.

- Why is there a significance in chemotherapy while the number re the same in the table S1?

R: Thank you for your observation. After reviewing the data, it appears that the significance regarding chemotherapy is indeed an error. We will correct the analysis and update the table accordingly.

- Line 201: please make clear what this means? Why drawback?

R: There is currently controversy regarding the oncological safety of using the uterine manipulator in endometrial cancer, especially due to complications associated with its use, such as uterine perforation, which could increase the risk of tumor dissemination. In our case, since this is a multicenter study, few centers that recruited patients used the manipulator in laparoscopic surgeries. Therefore, if its involvement in survival were to be demonstrated, it could represent a bias.

"Similarly, although the uterine manipulator is considered useful in laparoscopic surgery, its use remains a controversial topic due to the potential risks of uterine perforation, which could increase the likelihood of tumor dissemination. In robotic-assisted laparoscopic surgery (RAL), the use of a uterine manipulator is often not necessary, given the advantages that the robotic technique offers, such as enhanced visualization and precision. However, this is not the case in traditional laparoscopic approaches, where many centers continue to rely on the manipulator. In the context of our study, the uterine manipulator was employed in a limited number of centers. While some studies suggest that the use of the manipulator may negatively impact oncologic outcomes, there is currently no solid evidence or formal recommendation in the main guidelines for the management of endometrial cancer to discontinue its use. Although it was used in a few centers in our study, this could represent a potential limitation, as it may introduce bias if its involvement in survival outcomes is proven."

- the possible advantages of RAS could be discussed in more detail: obesity, elderly, ICG use, etc. 

R: Thank you for your suggestion. I will incorporate a more detailed discussion of the potential advantages of RAL.

"RAL safety and feasibility have been thoroughly investigated, with key advantages over standard LPS, including the 3-dimensional view, enhanced dexterity of the robotic arms, and a shorter learning curve. Some studies have compared both approaches using objective measurement tools, with shorter knotting and suturing times observed in robotic surgery compared to laparoscopy. This aspect is particularly important in overcoming the technical challenges posed by complex patients, such as those with endometrial cancer (EC). RAL offers a significant advantage in challenging surgical cases, such as those involving obese or elderly patients, both of which are factors commonly associated with endometrial cancer. The enhanced visualization and improved dexterity for suturing in confined spaces, such as those encountered in obese patients, provide a clear benefit of RAL over LPS. RAL has shown advantages in treating obese women with endometrial cancer, including reduced blood loss and lower rates of conversion to laparotomy. While some reports suggest that RAL may outperform LPS in terms of perioperative outcomes, the literature, though extensively studied, remains controversial on this issue. Additionally, RAL is associated with fewer postoperative complications, shorter hospital stays, and the reduced complexity of the approach, which greatly benefits frail patients, such as the elderly."

- if you discuss spillage, you also could add the aspect of tissue extraction and discuss differences between laparoscopy and RAS. 

Thank you for your insightful comment.

"Another important consideration is the need for protective maneuvers to prevent tumor spillage, including tumor rupture or morcellation. The advantages of RAL, compared to conventional laparoscopy (LPS), favor the implementation of protective measures that reduce the risk of tumor dissemination. Morcellation is easier, and tumor rupture is less likely due to the 3D visualization and the enhanced dexterity of the robotic arms, which allow for greater control and precision of movements."

Reviewer 3 Report

Comments and Suggestions for Authors

This is a relatively large retrospective analysis on the prognostic impact of RAL vs LPS on patients with early stage EC in four tertiary cancer center in Spain. 

The method of this retrospective analysis (propensity score analysis) was adequate.

The results confirm other published data on this issue, but the size of the cohort, the statistical method and the multicentric character of the study strengthen the relevance of this paper.

I have no major comments to this work and support the publication of the manuscript as it is.

Author Response

C- This is a relatively large retrospective analysis on the prognostic impact of RAL vs LPS on patients with early-stage EC in four tertiary cancer centers in Spain.

The method of this retrospective analysis (propensity score analysis) was adequate.

The results confirm other published data on this issue, but the size of the cohort, the statistical method and the multicentric character of the study strengthen the relevance of this paper.

I have no major comments to this work and support the publication of the manuscript as it is. 

R: Thank you for your thoughtful and supportive feedback. We appreciate your positive assessment of the study's design and findings. Your comments reinforce the relevance of our work, and we are grateful for your support in the publication of the manuscript.

Round 2

Reviewer 2 Report

Comments and Suggestions for Authors

Dear authors, thank you for providing the revised version of the manuscript. 

I have some additional comments: 

The advantages of RAL, compared to conventional laparoscopy (LPS), favor the implementation of protective measures that reduce the risk of tumor dissemination. Morcellation is easier, and tumor rupture is less likely due to the 3D visualization and the enhanced dexterity of the robotic arms, which allow for greater control and precision of movements."

Is there an advantage of RAS in tissue extraction? Isn't this the same problem in CLS and RAS? And if you argue that RAS is advantageous please cite the references.

"RAL safety and feasibility have been thoroughly investigated, with key advantages over standard LPS, including the 3-dimensional view, enhanced dexterity of the robotic arms, and a shorter learning curve. Some studies have compared both approaches using objective measurement tools, with shorter knotting and suturing times observed in robotic surgery compared to laparoscopy. This aspect is particularly important in overcoming the technical challenges posed by complex patients, such as those with endometrial cancer (EC). RAL offers a significant advantage in challenging surgical cases, such as those involving obese or elderly patients, both of which are factors commonly associated with endometrial cancer. The enhanced visualization and improved dexterity for suturing in confined spaces, such as those encountered in obese patients, provide a clear benefit of RAL over LPS. RAL has shown advantages in treating obese women with endometrial cancer, including reduced blood loss and lower rates of conversion to laparotomy. While some reports suggest that RAL may outperform LPS in terms of perioperative outcomes, the literature, though extensively studied, remains controversial on this issue. Additionally, RAL is associated with fewer postoperative complications, shorter hospital stays, and the reduced complexity of the approach, which greatly benefits frail patients, such as the elderly."

Please add references!!! There are some relevant papers on this topic comparing RAS and CLS.

Best regards.

Author Response

Thank you for your valuable feedback and suggestions.   

Is there an advantage of RAS in tissue extraction? Isn't this the same problem in CLS and RAS? And if you argue that RAS is advantageous please cite the references.  

R: Another important consideration is the need for protective maneuvers to prevent tumor spillage, including tumor rupture or morcellation [13,31]. While robotic-assisted laparoscopy (RAL) offers advantages such as 3D visualization and the enhanced dexterity of robotic arms, features that allow for greater control and precision during movements and could theoretically reduce the risk of secondary tumor dissemination during tissue extraction, there is currently insufficient evidence to demonstrate significant differences between RAL and conventional laparoscopy (LPS) in this regard [27].   13. Concin, N.; Matias-Guiu, X.; Vergote, I.; Cibula, D.; Mirza, M.R.; Marnitz, S.; Ledermann, J.; Bosse, T.; Chargari, C.; Fagotti, A.; et al. ESGO/ESTRO/ESP Guidelines for the Management of Patients with Endometrial Carcinoma. International Journal of Gynecologic Cancer 2021,31, 12–39, doi:10.1136/ijgc-2020-002230.   31.  Concin, N.; Planchamp, F.; Abu-Rustum, N.R.; Ataseven, B.; Cibula, D.; Fagotti, A.; Fotopoulou, C.; Knapp, P.; Marth, C.; Morice, P.; et al. European Society of Gynaecological Oncology Quality Indicators for the Surgical Treatment of Endometrial Carcinoma. International Journal of Gynecological Cancer 2021, 31, ijgc-2021-003178, doi:10.1136/ijgc-2021-003178.   27.  Coronado Martín, P.J.; Gracia, M.; Ramirez Mena, M.; Bellón del Amo, M.; García-Santos, J.; Fasero Laiz, M. The Well-Being of the Gynecological Surgeon Improves with the Robot-Assisted Surgery. ANALES RANM 2023, 139, 294–302, doi:10.32440/ar.2022.139.03.rev10.

Please add references!!! There are some relevant papers on this topic comparing RAS and CLS.

R: RAL safety and feasibility have been thoroughly investigated, with key advantages over standard LPS, including the 3-dimensional view, enhanced dexterity of the robotic arms, and a shorter learning curve [26]. Some studies have compared both approaches using objective measurement tools, with shorter knotting and suturing times observed in robotic surgery compared to laparoscopy [27]. This aspect is particularly important in overcoming the technical challenges posed by complex patients, such as those with endometrial cancer (EC) [28].

RAL offers a significant advantage in challenging surgical cases, such as those involving obese or elderly patients, both of which are factors commonly associated with endometrial cancer. The enhanced visualization and improved dexterity for suturing in confined spaces, such as those encountered in obese patients, provide a clear benefit of RAL over LPS [27]. RAL has shown advantages in treating obese women with endometrial cancer, including reduced blood loss and lower rates of conversion to laparotomy [29]. While some reports suggest that RAL may outperform LPS in terms of perioperative outcomes [30], the literature, though extensively studied, remains controversial on this issue [11]. Additionally, RAL is associated with fewer postoperative complications, shorter hospital stays, and the reduced complexity of the approach, which greatly benefits frail patients, such as the elderly [30].

11. Coronado, P.J.; Rychlik, A.; Baquedano, L.; García-Pineda, V.; Martínez-Maestre, M.A.; Querleu, D.; Zapardiel, I. Survival Analysis in Endometrial Carcinomas by Type of Surgical Approach: A Matched-Pair Study. Cancers (Basel) 2022, 14, 1081, doi:10.3390/cancers14041081.

26. Mäenpää, M.M.; Nieminen, K.; Tomás, E.I.; Laurila, M.; Luukkaala, T.H.; Mäenpää, J.U. Robotic-Assisted vs Traditional Laparoscopic Surgery for Endometrial Cancer: A Randomized Controlled Trial. Am J Obstet Gynecol 2016, 215, 588.e1-588.e7, doi:10.1016/j.ajog.2016.06.005.

27. Coronado Martín, P.J.; Gracia, M.; Ramirez Mena, M.; Bellón del Amo, M.; García-Santos, J.; Fasero Laiz, M. The Well-Being of the Gynecological Surgeon Improves with the Robot-Assisted Surgery. ANALES RANM 2023, 139, 294–302, doi:10.32440/ar.2022.139.03.rev10

28. Corrado, G.; Vizza, E.; Cela, V.; Mereu, L.; Bogliolo, S.; Legge, F.; Ciccarone, F.; Mancini, E.; Gallotta, V.; Baiocco, E.; et al. Laparoscopic versus Robotic Hysterectomy in Obese and Extremely Obese Patients with Endometrial Cancer: A Multi-Institutional Analysis. European Journal of Surgical Oncology 2018, 44, 1935–1941, doi:10.1016/j.ejso.2018.08.021.

29. Gracia, M.; García‐Santos, J.; Ramirez, M.; Bellón, M.; Herraiz, M.A.; Coronado, P.J. Value of Robotic Surgery in Endometrial Cancer by Body Mass Index. International Journal of Gynecology & Obstetrics 2020, 150, 398–405, doi:10.1002/ijgo.13258.

30. Coronado, P.J.; Herraiz, M.A.; Magrina, J.F.; Fasero, M.; Vidart, J.A. Comparison of Perioperative Outcomes and Cost of Robotic-Assisted Laparoscopy, Laparoscopy and Laparotomy for Endometrial Cancer. European Journal of Obstetrics & Gynecology and Reproductive Biology 2012, 165, 289–294, doi:10.1016/j.ejogrb.2012.07.006.